# Environmental Knowledge of Participants' Outdoor and Indoor Physical Education Lessons as an Example of Implementing Sustainable Development Strategies

Marcin Pasek [1,*], Elena Bendíková [2], Michalina Kuska [3], Hanna Żukowska [3], Remigiusz Dróżdż [1], Dariusz Jacek Olszewski-Strzyżowski [1], Magdalena Zając [4] and Mirosława Szark-Eckardt [3]

1  Faculty of Physical Culture, Gdansk University of Physical Education and Sport, 80-336 Gdansk, Poland; remik.pit@gmail.com (R.D.); jacek.olszewski63@gmail.com (D.J.O.-S.)
2  Faculty of Arts, Matej Bel University, 974 01 Banská Bystrica, Slovakia; Elena.Bendikova@umb.sk
3  Institute of Physical Education, Kazimierz Wielki University, 85-064 Bydgoszcz, Poland; michalinakuska@ukw.edu.pl (M.K.); zukowska@ukw.edu.pl (H.Ż.); szark@ukw.edu.pl (M.S.-E.)
4  Department of Special Pedagogy and Speech Therapy, Kazimierz Wielki University, 85-064 Bydgoszcz, Poland; magzaj@ukw.edu.pl
*  Correspondence: marcin.pasek@awf.gda.pl

**Abstract:** (1) Background: The purpose of the study was to assess the impact of physical activity outdoors in nature as part of physical education in schools on the level of knowledge and ecological attitudes. (2) Material and methods: A total of 220 students took part in the study, with 103 of them in the treatment group, which usually practiced outdoor physical education classes, and 117 in the control group, which practiced mainly indoor. The project lasted 21 months, covering the last two years of primary school. The authors used the Children's Environmental Attitude and Knowledge Scale CHEAKS in this study. The authors sought for an answer to the question of whether bringing a young person closer to nature by participating in a greater number of outdoor physical education lessons results in in-depth environmental knowledge. (3) Results: The appearance of seven statistically significant differences in ecological knowledge in the final study in favor of the group having outdoor physical education lessons proves the cognitively and visually stimulating role of a natural environment for physically active people. The location of physical education lessons turned out to be a much stronger condition for in-depth knowledge than gender, place of residence, parents' education level, and subjective assessment of the financial satisfaction level. (4) Conclusion: These results are an incentive to further developing the young generation's contact with nature through outdoor physical education lessons.

**Keywords:** ecological knowledge; outdoor physical education lessons; sustainable development

## 1. Introduction

The phenomenon of modern humans moving away from nature has been widespread and signaled since the 1970s [1], becoming a root cause of unsustainability [2]. Recently, there has been a growing number of calls for human reconnection with nature [3] as a system of links with sustainable behaviors [4], leading to sustainability [5]. However, most calls to reconnect humans with their environment are speculative and vague [6]. In contrast, it is undeniable that humanity needs to reduce its distance from the natural world by learning more about it, while increasing its distance in the sense of direct human impact on ecosystems [7]. This is currently difficult due to people living in increasingly dense metropolitan areas [8]. Unfortunately, those who experience this condition in their youth are likely to transfer this particular ignorance to their own children, leading to a persistent civilization trend [9]. Meanwhile, as these children lose their freedom to roam around their natural surroundings, they also lose the need to do so over time, becoming a generation of

confined spaces [10] and allocating more and more time to television, computer games, and social media [11]. It is comforting to know, however, that youth who maintain interactions with nature remain in this close relationship with nature well into adulthood [12,13]. This serves an incentive to implement environmental strategies in teaching. This allows teenagers to focus on contact with nature as part of their lessons at school, which is often lacking at their age. Nature is an unconscious value for representatives of this age group, unlike children at younger ages [14]. This is consistent with research in educational psychology on activity development during childhood and adolescence [15]. However, as part of environmental education, there is value in identifying and supporting the needs of youth to connect with nature [16]. This should assist them in achieving a fuller identification with sustainability issues.

The European Union's Sustainable Development Strategy is a multi-faceted approach to shaping policies for the coexistence of humans and their environment. It was formulated in detail in the document Sustainable Development Agenda: 2030, containing 17 goals of key importance for human development. Targets to be achieved by 2030 are grouped in five areas (the so-called 5xP): people, planet, prosperity, peace, and partnership. The targets cover a wide range of challenges such as poverty, famine, health, gender equality, climate change, sustainable development, peace and social justice, and education. The latter aspect has been thoroughly analyzed within the framework of Target 4 of the Agenda [17]. It is high-quality education aimed at eliminating the existing barriers on the way to effective school formation of the young generation. These include the shortage of adequately trained teachers, unsatisfactory conditions in schools, and limited access to schools for children from rural areas. In this context, it is possible to talk about the approximation of the assumptions of sustainable development and physical education in contact with nature. Extending the area of influence on the student during PE lessons by including outdoor activities may provide the basis for developing new competences of teachers in terms of knowledge and skills. Field-based physical education lessons can also be a response to the lack of adequate school infrastructure. Apart from that, this type of class also enables the elimination of differences between rural and municipal areas related to students' access to this infrastructure.

The current state of understanding the issues in shaping ecological knowledge and attitudes in contact with the natural environment allows us to state that outdoor physical education lessons may become an effective method of environmental formation in the school education system.

The phenomenon of sustainable development as defined in 1987 initially included only issues related to the economic activity of individual societies [17]. However, the end of the first decade of this century brought interest in non-economic aspects of this phenomenon. This has resulted in research in culture and education. Sustainable development has become one of the fundamental concerns of many countries with high socioeconomic potential but with internal differences for various reasons. Examples include Italy, with its north–south differences; Germany, with its still perceived differences between the former East and West German Länder; or Spain, known for its urban–rural differences [18].

It also had to become one of the fundamental socio-political aspects of the European Union. The supra-state and supranational nature of this structure, combined with its main objectives—integration, improvement, and unification of living conditions and free movement of people—require the implementation of a policy to balance the development of Member States. It is worth emphasizing that not only the economic differences but also the differences in access to culture, education, health care, and modern health-promoting lifestyles are noticed. A very important area requiring efforts to unify the quality of social life is certainly physical culture and the educational process introducing young generations to its values.

Environmental education is a concept of training and educating society in the spirit of respect for the natural environment. Environmental education is guided by one main idea that can be summarized in these words: "think globally, act locally", which can be

identified with highly developed ecological awareness. This awareness translates into specific attitudes [19], which are a tendency for a psychological perception of the natural environment [20]. When analyzing the shaping of attitudes in children, it can be noticed that this process usually begins with the formation of actions or emotions, and the cognitive aspect of the attitude develops latest [21]. For this reason, there is an obvious need for the school to support the family in the harmonious development of the child.

Environmental education includes introducing topics concerning environmental protection and nature protection to school programs, enabling the combination of environmental knowledge with a humanistic attitude as part of general school and extracurricular education. Out-of-school education can be understood in two ways: first, as the implementation of didactic tasks outside school and, second, as education realized by out-of-school, specialized, both public and non-public, and educational institutions as well as by numerous environmental organizations. The preparation of ecological programs, which are specific educational tools [22] at all possible education levels, should significantly enhance the effectiveness of education [23]. In addition to current goals of this strategy, there is also an increasing interest in the fate of future generations, for which modern humans should feel responsible to a large extent [24,25].

In connection with shaping the desired environmental attitudes, educational efforts are also directed at developing theoretical knowledge and its practical use. Knowledge about the environment has an interdisciplinary character aimed at the development of a human being aware of the need to solve current environmental problems [26]. One can distinguish three forms of ecological knowledge: systematic knowledge, knowledge about activities, and knowledge about the environment translating into effective environmental behaviors [27]. Sharing this knowledge within environmental courses leads to an increase in social responsibility for the issues of sustainable development [28,29] and to a growing belief in the possibility of personal contribution to environmental solutions [30,31]. In order to address the challenges posed by nature conservation; the pollution of soil, water, and air; deforestation; salinity; urbanization; global warming; and other climate changes, it is essential to deeply explore knowledge resources for representatives of the young generation with a view of developing the desired skills and ecological attitudes in young people [32].

Physical activity in open spaces comprises a basis for many domestic health programs for the adult society, but to be implemented in adult life, physical activity patterns are realized throughout stages of school education to a large extent. New Zealand has the longest learning experience outside the classroom. In 1849, R. Huntley founded a boys' school offering outdoor activities. After one hundred years, its assumptions were restored on the initiative of L.B. Sharp, who claimed that education begins behind school doors, where one can find things described and presented in textbooks. He also suggested that the school building should be treated as a command staff that manages outdoor activities [33]. Attention is drawn to a variety of offered activities including tourism and recreation, sport including survival training, natural and ethnographic activities, and those relating to art and drama. The main objectives of this education involve enriching lessons with adventure and joy that comes from it, supporting individual development, supporting interpersonal relationships, developing sensitivity, and learning utilitarian behaviors [34].

In Australia, following the models of its neighbors and intending to safeguard the implementation of outdoor education in the future, staff training is organized. Chevalier College in New South Wales offers courses for teachers and students to prepare them for work involving outdoor education with children aged eight or over. Courses with names such as Wilderness Leadership, Wilderness Studies, or Wilderness Expeditioning have both a theoretical and practical nature. The latter group includes night field trips, climbing mountain peaks, crossing rivers and streams, fast water canoeing, sailing, long-hour hiking, and finally preparing a camp for several nights on your own, among others [35].

Outdoor activities in the Oceanic zone are rather therapeutic in view of the civilization impact, providing a natural school of life in difficult conditions, to a greater extent. Contrarily, English patterns of outdoor and adventure education, despite the elements

of learning about and protecting nature, mainly appeal to a reduction in negative effects of the contemporary hectic life [36–40]. This education covers three areas: 1. outdoor pursuits—a variety of physical adventure activities designed to develop resourcefulness in hard times; 2. outdoor studies—education forms connected with culture, architecture, civic knowledge, and nature, based on observation and adaptation to changing external conditions; and 3. residential elements—learning cooperation and shaping attitudes towards overcoming difficulties together. Physical education takes about 10 percent of learning time and is enriched with the outdoor education mentioned above, which is an opportunity for integrated learning of many subjects at the same time. At the end of the last century, regional outdoor educational centers were even established facilitating the organization of the so-called outdoor week [41].

The experiences of the Germans are based on combining field activities with school sports and environmental education. Known for their order and discipline, they are heading towards the construction of sports and recreational facilities offering close contact with nature located near urban areas. Recreational and adventure activities are primarily of a sports nature, while the experience of the natural environment is safeguarded by detailed safety and security regulations. Theoretical preparation is followed by weekly school camps, supplemented by different trips and conflict-free living with nature lessons. Students develop, among other things, the ability of quiet paddling, silent communication, or discreet nature watching [42].

In 1991, in France, a decision was made to pilot a combination of outdoor activities with traditional physical education lessons. Eight specialties of outdoor lessons were approved: alpine climbing, outdoor orienteering, canoeing, cross-country cycling, downhill and cross-country skiing, sailing, and oxygen scuba diving. To illustrate, the school board in Rouen in Normandy recommends a year-round program consisting of seven 90-min lessons for beginners and fourteen 90-min lessons for advanced users [43,44].

The overall health goals guide American outdoor-environmental education. Its programs are sponsored by schools, youth organizations, churches, nature centers, and private individuals with access to city parks, forests, and municipal parks. The need to acquire managerial skills in this area dates back to the 1970s [45], but its practical implementation through courses for health educators has a slightly shorter history [46–48]. It takes place even in the far north of the USA and in Canada, far away from destructive civilization influences [49,50]. During long winters, students go camping, ice skating, sledging, snow carving, winter orienteering, adventuring in deep snow, and ice-hole fishing.

Contrary to the abovementioned positive examples, in Polish reality, supporting the development of ecological competences is assessed by teachers as one of the less important tasks of physical education [51]. The low rank of these competences may result from the fact that combining thinking about physical education with reflection on environmental education does not enjoy long tradition in Poland, the enrichment of which depends to a large extent to engagement in educational projects connected with the natural environment. A partial response to this need was educational activities arranged by Pańczyk [52], who took up an analysis of the biological, health, and educational effects of physical education in nature and in the gym. The findings proved the greater effects of lessons in open spaces, which made it possible to formulate a postulate to increase the volume of the classes of this type in physical education programs.

The COVID-19 pandemic, which has accompanied humans for many months, forces a deeper understanding of the relationship between humans and the natural environment. This need has been expressed many times in recent times [53–56]. It stems from the fact that, in the process of physical education, there are not many alternatives for the student to obtain a closer at the world with its surrounding nature.

The problem of the undertaken research concerns the effectiveness of the school physical education system. The authors suggest the need for more field-based physical education lessons. They see it as a tool to improve school strategies referring to sustainable development. The results of surveys conducted in Germany indicate that, in general, students

and teachers are satisfied with the aesthetics and functionality of school grounds [15]. However, these are subjective perceptions, resulting from the fact that most are unaware of the possibilities of a truly well-designed schoolyard. Meanwhile, there are ideas for using the schoolyard as a successful teaching space for sustainable education [57]. Investment shortfalls are not the only barrier that can discourage outdoor physical education. Additionally, highlighted by teachers is a sense of unpreparedness and lack of confidence in outdoor teaching [58], concerns about classroom management and children's safety in the outdoors [59], lack of financial support for this type of project [60], or finally the recognition of the support of environmental competence as one of the less important tasks of PE [51]. Trouble can also arise when signing year-round parental consent forms for their children to be outdoors and when determining the minimum amount of time that classes should spend each week learning outdoors [61].

Due to the above limitations, it is difficult in the Polish reality to convince teachers to move part of the educational process to the field. However, education for sustainable development has been included in school programs, including education and prevention programs. In the core curriculum, one can find references to education in the context of acceptance for other people, shaping of attitudes of respect for the natural environment, including the dissemination of knowledge on the principles of sustainable development [62]. Knowledge as an educational competence resulting from participation in physical education is included in the broad discussion because of the many content contexts used in the lessons [63]. In fact, concerns sometimes arise that current forms of physical education teacher education do not provide the tools necessary to work in light of the challenges of contemporary physical education [64]. However, physical activity in contact with nature allows us, in this case, to create a successful perspective.

Whether the experience of contact with nature fosters learning has been conjectured about until recently without the support of scientific evidence. Currently, however, there is a steady increase in work supporting the fact that experiencing nature stimulates cognitive processes as well as personal development and environmental management [65]. Evidence suggests that exposure to nature increases cognitive abilities among students through improved attention [66], stress reduction [67], discipline [68], engagement [69], and increased physical activity and improved fitness [70,71]. Students randomly assigned to classrooms with a green view performed better on concentration tests than those assigned to views of the transformed environment or classrooms without windows [72]. The positive effects of nature on attention have been observed using neurocognitive tests [73] and by studying students on field trips [74].

The aim of the study was to assess the impact of physical education lessons in the outdoor and indoor formula on the level of ecological knowledge. For correct statistical verification, a research hypothesis was constructed:

**Hypothesis 1 (H1).** *The different location of physical education including outdoor and indoor lessons is related to the extent of knowledge acquired such that those taking part in outdoor lessons have a higher level of knowledge than those taking part in indoor activities.*

## 2. Materials and Methods

The study covered four schools in the Pomeranian Voivodeship in the northern part of Poland. The study involved 220 students who formed two groups: treatment and control. Characteristics of the study group are presented in Table 1.

**Table 1.** Characteristics of the study group.

| Gender | Treatment Group (*n* = 103) | Control Group (*n* = 117) |
|---|---|---|
| Male | 49 | 63 |
| Female | 54 | 54 |

The study period lasted two years and involved the fifth and sixth forms of primary school. The experimental group subjects were 11.26 years old ($\pm$0.32) during the initial test, and the control group individuals were 11.28 years ($\pm$0.32). During the final test, the average ages of experimental group subjects were 12.96 years ($\pm$0.32), and those in the control group were 12.98 years ($\pm$0.32). The same pupils participated in the two-year project covering four terms in the fifth and fourth forms. Material supervision over the classes throughout the duration of the project was exercised by ten teachers. In two out of eight cases, after the first year of cooperation with the class, the teacher of the group was changed for reasons unrelated to the experiment. All of them had full university educations in physical education. In the schools involved in the study, they conducted classes according to the same didactic and educational plan, which contained all of the basic curriculum content and took into account its prospective realization both in open spaces and inside school. The initial and realized assumption was that, in each school, classes were conducted by the same teacher in the experimental and control groups.

The study used the scheme of an educational experiment. The usefulness of this method in relation to pedagogical analyses has been repeatedly recognized so far [75–77]. The scheme is based on the evaluation of the phenomenon under normal conditions, allowing the modification of existing conditions by the researcher [78]. The essence of the method is the selection of experimental and control groups that show the range of changes under the influence of a specific variable. The generally applicable rule is the most far-reaching selection of both groups obtained for variables such as age, number of subjects, the level of their biological development, as well as environmental circumstances [79]. The principle according to which this research was conducted were J.S. Mill's canons of the only difference [80] between the experimental and control groups. According to this principle, the case in which the studied phenomenon occurs and the case in which it does not occur have all circumstances in common except one, which is present only in the first case. The only differentiating factor in these studies was the different number of hours of outdoor physical education classes in the experimental and control groups. As part of projects of this type, a certain number of people are subjected to the same measurement twice over a period of time. In this case, environmental attitudes and knowledge are measured. By comparing the results of the initial and final study, the researcher is able to identify changes and to determine the dynamics of this variability [81]. In the classical scheme of parallel groups used, after the initial measurement, an independent variable otherwise known as an experimental factor was introduced in the experimental group, and after less than two years, the measurement was repeated. However, in the control group, no experimental factor was introduced after the first measurement, but after the same time, the measurement was also repeated. Conclusions concerning the influence of the independent variable were drawn from the existing difference between initial and final measurement values.

It was assumed that experimental group subjects would take part in a significantly bigger number of outdoor physical education lessons than their peers from control groups. Initially, the authors followed the earlier research assumptions of Pańczyk [52] who had set the number of outdoor lessons in the experimental groups at 75% and in the control groups at 25–33%. In our study, the average level of 60–65% was eventually achieved in the experimental classes and 30% in the control classes.

The study of ecological knowledge and attitudes applied the Children's Environmental Attitude and Knowledge Scale CHEAKS by F. Leeming et al. [82]. Components of ecological knowledge and attitudes were measured on a Likert scale based on the earlier assumptions of Ogunjinmi et al. [83], with very false = 1, mostly false = 2, not sure = 3, mostly true = 4, and very true = 5.

In one point connected to knowledge, a necessary correction was made to the text in relation to the American original [82] and its Nigerian version [83]. It consisted of replacing the countries the USA/Nigeria with Poland in the questionnaire.

The ecological knowledge scale consisted of thirty statements, which were divided into six categories: animals, pollution, general issues, water, energy, and recycling. Each of the categories consisted of five statements.

Before the research, the following assumptions were made:

1. Increased contact with nature lasting two years during physical education lessons is sufficient to improve the indicators of ecological knowledge and attitudes.

2. In addition to forms of physical activity outside the classroom with extended knowledge and in-depth ecological attitudes, it seems that the higher education of the parents of the studied pupils and perhaps the place of residence also seem to be decisive. According to previous observations [84], the industrial environment that does not take into account the harmonious whole of human needs induces deprivation of the need to experience beauty and order. Under these circumstances, identification with an ecologically transformed environment may lead to low self-esteem and malaise.

The central limit theorem and the observation that the summary questionnaire met the assumption of normality of distribution (K-S d = 0.07615, $p < 0.20$) were used in test selection. At the same time, due to the Likert scale used, it was impossible to obtain normal distributions for the items on such a short response scale. However, 80% of the items (24 out of 30) had a distribution shape close to the Gauss curve. The choice of parametric statistics was dictated by three considerations: the use of parametric test in the analysis of differences, the Likert scale characteristics (for all items, Me = 3 was obtained), the use of analogous measures in earlier publications of other authors (which allows for better correspondence of data) and the more common use of the statistics used in published research.

In statistical analysis, parametric descriptive statistics (Mean, SD) were applied to characterize variables. Student's *t*-test was used to study the differences between the means of the compared groups and Pearson's r to study the relationships between variables (linear correlation coefficient). Linear regression analysis of the explained variables was also conducted. The results that met the condition of $p < 0.05$ were considered statistically significant.

The research proposal was approved by the authorities of the Academy of Physical Education and Sport in Gdańsk. The Ethics Committee, represented by the rector, acting on the basis of the Regulation of the Minister of National Education and Sport of 9 April 2002, on the conditions for conducting experimental activities by schools and public institutions, concluded an agreement with the Pomeranian education superintendent on a project assessing the quality of physical education in schools in the Pomorskie voivodeship (project no. 17/03/05).

Consent to conduct research in schools was obtained from the school principals. Prior to the study, consent was also obtained from parents or legal guardians of the children. The participants decided for themselves whether they wanted to take part in the study. Participation in the research was voluntary. Anonymity and confidentiality were highlighted throughout the study. The data were kept in a closed and safe place and accessed only by scientists and a statistician.

## 3. Results

The total score for all responses at the final and initial measurements in the treatment and control groups allowed us to proceed with cautious optimism to verify the research hypothesis.

The comparison of the results obtained from the ecological knowledge study carried out on the experimental and control group in the final study showed statistically significant differences in favor of the experimental group in the case of statements 4 and 5 in the field of environmental pollution. The most significant differences, also in favor of the experimental group subjects, were found in the study of knowledge on general issues. This applied to theorems 1, 3, 4, and 5. Students exercising outside also showed significantly greater concern related to the allocation of large spaces to landfills (theorem 4 in the field of recycling knowledge). Overall, out of thirty statements checking the knowledge level, in seven cases, the results were better in the experimental group. Summed score for all responses at the final and initial measurements in the treatment and control groups is presented in Table 2. The results of the initial and final research on ecological knowledge are presented in Tables 3 and 4.

**Table 2.** Summed score for all responses at the final and initial measurements in the treatment and control groups.

| Variable | Treatment Group | Control Group | *t* | df | *p* |
|---|---|---|---|---|---|
| Knowledge final measurement | 89.77 | 87.25 | 4.00 | 218 | 0.000 |
| Knowledge initial measurement | 87.25 | 87.52 | 0.42 | 218 | 0.673 |

**Table 3.** Comparison of the results of the ecological knowledge study in the experimental and control groups in the initial measurement.

| Knowledge Statement | Treatment Group | Control Group | *t* | *p* |
|---|---|---|---|---|
| | Mean $\pm$ SD | Mean $\pm$ SD | | |
| Item 1: Animal knowledge | | | | |
| 1 Most elephants are killed every year to provide people with ivory | 3.04 $\pm$ 0.86 | 2.96 $\pm$ 0.89 | −0.69 | 0.489 |
| 2 Catching tuna in the ocean also kills many dolphins | 2.84 $\pm$ 0.66 | 2.94 $\pm$ 0.79 | 0.96 | 0.337 |
| 3 Animals alive today are most likely to become extinct in the nearest future | 2.95 $\pm$ 1.17 | 2.97 $\pm$ 1.04 | 0.15 | 0.878 |
| 4 Killing animals like wolves that eat others may increase the number of other animals | 2.84 $\pm$ 1.08 | 2.69 $\pm$ 0.83 | −1.17 | 0.240 |
| 5 A species that no longer exists is extinct | 3.06 $\pm$ 0.93 | 3.06 $\pm$ 1.07 | 0.00 | 0.997 |
| Item 2: Pollution knowledge | | | | |
| 1 The most pollution of water source is caused by chemical run off from farms | 3.04 $\pm$ 0.94 | 3.14 $\pm$ 1.01 | 0.72 | 0.467 |
| 2 Nitrates and phosphates are the most common poison found in water | 2.77 $\pm$ 1.03 | 2.94 $\pm$ 0.82 | 1.36 | 0.173 |
| 3 High octane gas does not do much to reduce the pollution by automobiles | 2.67 $\pm$ 0.94 | 2.63 $\pm$ 1.00 | −0.35 | 0.721 |
| 4 Most air pollution in our big cities comes from cars | 3.04 $\pm$ 0.66 | 3.07 $\pm$ 0.70 | 0.30 | 0.760 |
| 5 Most lead in our air is caused by cars | 2.89 $\pm$ 0.80 | 2.77 $\pm$ 1.05 | −0.90 | 0.366 |

**Table 3.** *Cont.*

| Knowledge Statement | Treatment Group | Control Group | t | p |
|---|---|---|---|---|
| | Mean ± SD | Mean ± SD | | |
| Item 3: Knowledge on general issues | | | | |
| 1 Ecology is the study of the relationship between organisms and their environment | 2.86 ± 0.84 | 2.91 ± 0.80 | 0.45 | 0.650 |
| 2 Overpopulation is dangerous to earth's environment | 2.83 ± 0.89 | 2.89 ± 0.85 | 0.52 | 0.597 |
| 3 I am worried about environmental problem | 2.90 ± 1.18 | 2.96 ± 0.76 | 0.47 | 0.636 |
| 4 Environmental problems are threats to all living things in the world | 3.01 ± 0.85 | 3.04 ± 0.97 | 0.18 | 0.851 |
| 5 Ecology assumed that man is related to other parts of nature | 2.99 ± 0.67 | 2.91 ± 0.74 | −0.78 | 0.435 |
| Item 4: Water knowledge | | | | |
| 1 Phosphates are harmful in the sea water because they suffocate fish by increasing algae | 2.80 ± 0.82 | 2.81 ± 0.86 | 0.05 | 0.957 |
| 2 Building dam on a river damages the river's natural ecosystem | 2.82 ± 1.22 | 2.95 ± 1.18 | 0.81 | 0.417 |
| 3 Sulphur dioxide is most responsible for creating acid rain | 2.87 ± 0.77 | 2.96 ± 0.95 | 0.77 | 0.437 |
| 4 Underground waters are found in aquifers | 2.93 ± 0.70 | 2.90 ± 0.77 | −0.25 | 0.795 |
| 5 The main problem with the use of aquifers for water supply is becoming used up | 2.86 ± 1.00 | 3.05 ± 0.87 | 1.47 | 0.141 |
| Item 5: Energy knowledge | | | | |
| 1 Burning coal for energy releases carbon dioxide and other pollutants into the air | 2.81 ± 0.93 | 2.91 ± 0.98 | 0.75 | 0.448 |
| 2 Solar is an example of perpetual energy source | 2.81 ± 0.81 | 2.81 ± 0.78 | −0.03 | 0.973 |
| 3 Coal and petroleum are examples of fossil fuels | 2.95 ± 0.64 | 2.87 ± 0.83 | −0.78 | 0.434 |
| 4 An example of non-renewable resources is petroleum | 2.95 ± 0.79 | 3.00 ± 1.02 | 0.45 | 0.647 |
| 5 Hot water heater uses the most energy in an average house in Poland | 3.04 ± 0.94 | 2.89 ± 0.82 | −1.26 | 0.206 |
| Item 6: Recycling knowledge | | | | |
| 1 Compared to other papers. recycled paper takes less energy to make | 2.92 ± 0.84 | 2.73 ± 0.87 | −1.60 | 0.109 |
| 2 Garbage is dumped from the garbage trucks to a landfill where it is buried | 2.96 ± 0.81 | 2.85 ± 1.07 | −0.81 | 0.414 |
| 3 Pre-cycling means that people buy things that can be used again | 2.95 ± 0.77 | 2.92 ± 0.85 | −0.25 | 0.797 |
| 4 The main problem with landfills is that it takes up too much space | 2.86 ± 0.91 | 2.84 ± 0.87 | −0.14 | 0.882 |
| 5 An item which cannot be recycled and used again is known as disposable diapers | 2.85 ± 0.90 | 3.00 ± 0.90 | 1.26 | 0.207 |

**Table 4.** Comparison of the results of the ecological knowledge study in the experimental and control groups in the final measurement.

| Knowledge Statement | Treatment Group | Control Group | $t$ | $p$ |
|---|---|---|---|---|
| | Mean $\pm$ SD | Mean $\pm$ SD | | |
| Item 1: Animal knowledge | | | | |
| 1 Most elephant are killed every year to provide people with ivory | 2.90 $\pm$ 0.83 | 2.82 $\pm$ 0.96 | −0.67 | 0.500 |
| 2 Catching tuna in the ocean also kills many dolphin | 2.89 $\pm$ 0.85 | 2.82 $\pm$ 0.81 | −0.57 | 0.568 |
| 3 Animals alive today are most likely to become extinct in the nearest future | 2.95 $\pm$ 0.78 | 2.96 $\pm$ 0.65 | 0.14 | 0.882 |
| 4 Killing animals like wolves that eat others may increase the number of other animals | 3.01 $\pm$ 0.98 | 2.94 $\pm$ 0.86 | −0.63 | 0.526 |
| 5 A species that no longer exist is extinct | 2.92 $\pm$ 0.77 | 2.99 $\pm$ 0.93 | 0.59 | 0.553 |
| Item 2: Pollution knowledge | | | | |
| 1 The most pollution of water source is caused by chemical run off from farms | 2.83 $\pm$ 0.87 | 2.88 $\pm$ 0.81 | 0.47 | 0.637 |
| 2 Nitrates and phosphates are the most common poison found in water | 3.00 $\pm$ 1.12 | 2.95 $\pm$ 0.89 | −0.38 | 0.700 |
| 3 High octane gas does not do much to reduce the pollution by automobiles | 3.00 $\pm$ 0.91 | 2.97 $\pm$ 0.82 | −0.30 | 0.763 |
| 4 Most air pollution in our big cities comes from cars | 3.17 $\pm$ 0.75 | 2.87 $\pm$ 0.91 | −2.65 | 0.008 |
| 5 Most lead in our air is caused by cars | 3.21 $\pm$ 0.93 | 2.88 $\pm$ 0.88 | −2.71 | 0.007 |
| Item 3: Knowledge on general issues | | | | |
| 1 Ecology is the study of the relationship between organisms and their environment | 3.37 $\pm$ 0.78 | 3.04 $\pm$ 0.85 | −3.02 | 0.002 |
| 2 Overpopulation is dangerous to earth's environment | 2.80 $\pm$ 0.80 | 2.87 $\pm$ 0.65 | 0.67 | 0.502 |
| 3 I am worried about environmental problem | 3.28 $\pm$ 0.80 | 2.95 $\pm$ 1.13 | −2.41 | 0.016 |
| 4 Environmental problems are threats to all living things in the world | 3.41 $\pm$ 0.79 | 2.86 $\pm$ 1.02 | −4.43 | 0.001 |
| 5 Ecology assumed that man is related to other parts of nature | 3.34 $\pm$ 0.76 | 3.08 $\pm$ 0.93 | −2.27 | 0.023 |
| Item 4: Water knowledge | | | | |
| 1 Phosphates are harmful in the sea water because they suffocate fish by increasing algae | 2.68 $\pm$ 0.88 | 2.88 $\pm$ 0.97 | 1.51 | 0.131 |
| 2 Building dam on a river damages the river's natural ecosystem | 2.79 $\pm$ 1.09 | 2.82 $\pm$ 1.01 | 0.17 | 0.864 |
| 3 Sulphur dioxide is most responsible for creating acid rain | 2.88 $\pm$ 0.85 | 2.73 $\pm$ 0.91 | −1.23 | 0.216 |
| 4 Underground waters are found in aquifers | 2.88 $\pm$ 0.86 | 3.00 $\pm$ 0.69 | 1.10 | 0.270 |
| 5 The main problem with the use of aquifers for water supply is becoming used up | 3.02 $\pm$ 0.92 | 2.86 $\pm$ 0.87 | −1.37 | 0.171 |

**Table 4.** *Cont.*

| Knowledge Statement | Treatment Group Mean ± SD | Control Group Mean ± SD | t | p |
|---|---|---|---|---|
| | | | | |
| Item 5: Energy knowledge | | | | |
| 1 Burning coal for energy releases carbon dioxide and other pollutants into the air | 2.95 ± 0.88 | 2.88 ± 0.84 | −0.60 | 0.543 |
| 2 Solar is an example of perpetual energy source | 2.93 ± 0.83 | 2.82 ± 0.87 | −0.89 | 0.373 |
| 3 Coal and petroleum are examples of fossil fuels | 2.94 ± 1.05 | 2.94 ± 1.14 | 0.04 | 0.962 |
| 4 An example of non-renewable resources is petroleum | 2.88 ± 0.91 | 3.01 ± 0.85 | 1.12 | 0.262 |
| 5 Hot water heater uses the most energy in an average house in Poland | 3.00 ± 1.04 | 3.05 ± 0.74 | 0.34 | 0.731 |
| Item 6: Recycling knowledge | | | | |
| 1 Compared to other papers. Recycled paper takes less energy to make | 2.92 ± 0.98 | 2.83 ± 0.84 | −0.68 | 0.492 |
| 2 Garbage is dumped from the garbage trucks to a landfill where it is buried | 2.94 ± 0.83 | 2.82 ± 1.21 | −0.85 | 0.396 |
| 3 Pre-cycling means that people buy things that can be used again | 2.66 ± 1.04 | 2.81 ± 0.79 | 1.21 | 0.223 |
| 4 The main problem with landfills is that it takes up too much space | 3.28 ± 0.74 | 2.92 ± 0.69 | −3.68 | 0.001 |
| 5 An item which cannot be recycled and used again is known as disposable diapers | 2.79 ± 1.01 | 2.88 ± 1.00 | 0.67 | 0.497 |

In the final study, among the analyzed independent variables (experimental/control group, gender, place of residence, parents' education level, and financial satisfaction level), only in the case of the first variable, a high value of the statistics was observed. The regression model explaining the knowledge variable through the impact of the experimental/control group variable explains more than 64% of the variance with the statistic of $t = 3.88$ ($p < 0.001$). Analysis of regression is presented in Table 5.

**Table 5.** Analysis of regression.

| N = 220 | Summary of the Regression of Dependent Variable: Knowledge 2 in Total R = 0.28453055 R² = 0.08095763 Adjusted R² = 0.05506912 F(6,213) = 3.1272 p < 0.00587 Std Error of Estimate: 4.6743 | | | | | |
|---|---|---|---|---|---|---|
| | b (Beta) | Std Error Beta | b | Std Error b | t (213) | p |
| Intercept | | | −146.697 | 138.8276 | −1.05668 | 0.291853 |
| Group | 0.256750 | 0.066081 | 2.469 | 0.6354 | 3.88538 | 0.000136 |
| Gender | −0.018671 | 0.066471 | −0.179 | 0.6379 | −0.28089 | 0.779064 |
| Location | 0.046550 | 0.072142 | 0.447 | 0.6927 | 0.64525 | 0.519457 |
| Mother's educ. level | −0.092826 | 0.081126 | −0.355 | 0.3100 | −1.14422 | 0.253818 |
| Father's educ. level | 0.089754 | 0.082136 | 0.335 | 0.3063 | 1.09274 | 0.275740 |
| Living conditions | −0.041625 | 0.068085 | −0.400 | 0.6535 | −0.61137 | 0.541607 |

The results of the regression analysis indicate that final knowledge is dependent on whether the respondents had outdoor or indoor activities. Other elements related to family demographics and education did not affect it. At the same time, it is important to note that all variables explain only 8% of the variation in knowledge, which means that there are many other variables of importance that affect knowledge.

## 4. Discussion

Despite the lack of strong links between physical education, including field physical education and sustainable development, it should be mentioned that the international forum has identified as many as seven sustainable development goals that relate to physical activity, sport, and physical education [85]. Additionally, while it may be debatable to acknowledge them all, it seems that, for at least three of the objectives, the relationships are indisputable. One of these, known as Good Health and Well-Being in Task 4, refers to reducing mortality by promoting mental health and well-being. Outdoor physical education can support this goal, because of the correlates of well-being both with physical activity [86,87] and with the natural environment [88]. Within the next objective, which is good quality education, the first task can already be implemented based on outdoor physical education. The quality of education is enhanced by contact with nature [89–96]. One of the goals of the 2030 Agenda is also climate action understood as the implementation of national strategies. One of them could be outdoor physical education, which means reducing the cost of maintaining sports facilities [97].

However, the role of sport and physical education has been lauded as a fundamental right for all at least since the publication of the International Charter on Physical Education and Sport (UNESCO) in 1978. In addition to these goals, Agenda 2030 recognizes sport as an important facilitator of sustainable development and peace, supporting the empowerment of women and youth and achieving the goals in social inclusion (Strategy on Education for Health and Well-Being). Although the presented results do not clearly indicate the relationship between the level of environmental knowledge and the implementation of sustainable strategies, a short discussion can be held on the indirect relationships between these categories. The current targets of the Sustainable Development Agenda: 2030 are an extensive continuation of the eight Millennium Development Goals. Efforts towards these goals included, inter alia, the integration of physical activity into education, health, development, and peace programs. This was reflected in designating 2005 as the International Year of Sport and Physical Education [98].

Currently, the United Nations Office on Sport for Development and Peace focuses its activities on deepening the relationship between physical activity and sustainable development. One of the pillars of the sustainable development policy, which is creating an attitude of respect and tolerance towards each other, is fully possible to implement within group forms of outdoor physical activity. It has been proven that exposure to nature lowers the social dominance orientation [99].

Apart from social inclusion, sustainable development goals coincide with physical activity goals also in the area of health and education [100]. It is probably one of the first examples of such a broad approach to development policy that it also covers physical culture-related issues [101]. Undeniably, the intention of supporters of bringing people closer to the world of nature through outdoor physical activity is to raise ecological awareness, which should first improve the processes of knowledge perception, and then to become visible in achieving a sustainable development friendly identity.

In light of the research results demonstrated in this paper, outdoor physical education lessons appear to be a crucial factor shaping environmental knowledge. The assessment of ecological knowledge in consideration of the location of physical education lessons, broken down by outdoor and indoor activities, has not been taken into account in previous studies. Other studies dealt with the environmental attitudes of students without analyzing the determinants of these attitudes [102], showing that 80% of students preparing for studies in Alexandria had a negative attitude to environmental issues whereas the remaining 20% represented indifference. The studies conducted by Ogunjinmi et al. [83] showed that the status of school, namely its public or private character ($\beta = -10.08$; $p < 0.05$), comprised the only determinant of students' environmental attitudes. Other individual factors, such as age, gender, education level, and the nature of the class, broken down by its scientific and artistic profile, were not related to the students' environmental attitudes.

The same study [83] suggests that gender (β = 0.18; $p < 0.05$) and nature of the class (β = 0.34; $p < 0.01$) accounted for determinants of students' ecological knowledge, and thus, they constituted 18% of diversity in the relation between respondents' individual factors and their ecological knowledge. In terms of gender, this is in line with the study results suggesting that gender is a factor strongly influencing acquisition of environmental knowledge [103]. The result obtained by Ogunjinmi et al. [83] regarding gender and nature of the class was inconsistent with observations made by Akomolafe [104].

In a Turkish research study, a pre-test and a post-test were carried out without the control group among 64 students (38 boys and 26 girls) participating in an ecological curriculum. Their knowledge was tested using the Natural Sciences Knowledge test consisting of fifteen double choice issues. The pre-test showed the value of correct answers on a minimum level of four and a maximum level of fourteen with an average result being 9.80. On the other hand, in the final study, the values were 6–15 and 10.62, respectively, which appeared to be a statistically significant improvement ($p < 0.05$) [105]. More than a dozen studies related to secondary [106] and primary school [107] students also show a better understanding of environmental issues and faster acquisition of knowledge in individuals having more contact with nature, with emphasis on more effective combining theory with practice in this case [108]. The knowledge–attitude–behavior model indicates that an increase in the knowledge level affects attitudes, which generate environmental behaviors. Consequently, both knowledge and attitudes often determine an increase in ecological responsibility as an observed effect of outdoor education programs [109].

## 5. Conclusions

The conclusion that emerges from the presented research results is that sustainable development should become part of education and, at the same time, education must be part of sustainable development. The school system is therefore meant to be a tool to support the embodiment of the philosophy of sustainable development. It should be delivered, among others, by physical education teachers who are competent as environmental educators. Additionally, by gaining ecological knowledge through contact with nature, students increase their potential as promoters of sustainable development in the future.

The emergence of differences in environmental knowledge and attitudes in the final study for the benefit of the group having physical education lessons outside classroom may indicate the cognitive and visual stimulating role of natural environment in engaging in outdoor forms of physical activity. In the study group, physical activity in the natural environment turned out to be a much stronger determinant of in-depth knowledge than other analyzed factors, such as gender, place of residence, parents' education level, and subjective assessment of financial satisfaction.

The new concept of general education, assuming a departure from the propaedeutic and encyclopedic model in favor of a more utilitarian one, prefers to provide students with such information and skills that will allow them, inter alia, to coexist with the surrounding natural environment. In this approach, physical education should be understood as a carrier of the imperative of work on oneself and mainly include activities which create an alternative attitude towards one's own health and physical fitness, especially in regular contact with nature. This environment is conducive to self-education based on students' own activity, strengthened by the emotional foundation that is provided by the specific charm of classes conducted outside school. Outdoor activities effectively prepare students for the proper organization of free time, shaping permanent health and leisure habits.

On an ecological trip, one can introduce new content provided for in the curriculum, develop and enrich information, and check skills in a specific activity. Generally, students respond to outdoor classes positively, lively, and enthusiastically. Children, especially in early school age, deeply experience leaving the classroom and outdoor activities and are not always aware of the fact they can also learn a lot on such trips. All of the messages that pupils obtain from a trip are associated with immediate emotional experiences. It is also known that experience-based knowledge is more complete than content repeated many

times in the classroom. Combining theoretical issues from various fields of knowledge with practical activities during physical education lessons may contribute to optimizing the processes of active acquisition and understanding information. Indeed, situations arranged between one or another physical exercise in school premises will never be sufficient for the creation of knowledge related to physical education. Failure to take this fact into account is one of the reasons why the postulates of intellectualizing physical education remain in the sphere of educational myths. In order to effectively stimulate the development of knowledge on various topics, an organizationally independent process from physical education lessons is needed.

This will allow for more effective shaping of environmental and health awareness of young people, for whom contact with nature would not be limited only to physical improvement but would also become a source of spiritual values, in which the integration of physical and ecological culture towards future culture of life can be seen.

**Author Contributions:** Conceptualization, M.P.; methodology, E.B. and M.Z.; software, R.D.; validation, E.B.; formal analysis, E.B. and M.Z.; investigation, M.P. and M.S.-E.; resources, R.D.; data curation, M.K.; writing—original draft preparation, M.P.; writing—review and editing, D.J.O.-S. and M.S.-E.; visualization, D.J.O.-S.; supervision, H.Ż.; project administration, M.K. and M.S.-E.; funding acquisition, H.Ż. All authors have read and agreed to the published version of the manuscript.

**Funding:** This research received no external funding.

**Institutional Review Board Statement:** The study was conducted according to the guidelines of the Declaration of Helsinki, and approved by the Institutional Review Board of GDANSK UNIVERSITY OF PHYSICAL EDUCATION AND SPORT acting on the basis of the Regulation of the Minister of National Education and Sport of 9 April 2002 on the conditions for conducting innovative and experimental activities by public schools and institutions, concluded an agreement with the Pomeranian school superintendent concerning the project of evaluation and stimulation of the quality of physical education in (PE) schools in the Pomorskie Voivodeship (Project Number 17/03/05).

**Informed Consent Statement:** Informed consent was obtained from all subjects involved in the study.

**Data Availability Statement:** Not applicable.

**Conflicts of Interest:** The authors declare no conflict of interest.

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
