# Peer review of "Environmental Knowledge of Participants’ Outdoor and Indoor Physical Education Lessons as an Example of Implementing Sustainable Development Strategies"

_sustainability, doi:10.3390/su14010544_

Round 1

Reviewer 1 Report

The paper deals with the environmental knowledge of the participants of outdoor or indoor physical education classes as an example to implement sustainable development strategies.

Although this topic seems to be of importance, also considering the sustainable development goals of the 2030 agenda, there are some revisions that I recommend before accepting this paper for publication especially in the materials and methods section.

Method:
- A table is suggested to characterize the sample and to understand it in a clearer way.
- I would like to read how the questionnaire items were constructed and validated. is there a confirmatory factor analysis? The description of the instrument needs to be improved.

It is suggested to include a specific section on statistics that justifies the choice of tests. 

It is recommended to perform the kolmogorov smirnov test to know if the data meet the normality assumption.

If they meet the assumption of normality, it is acceptable to perform the t-test, otherwise the data should be presented as Me and IQR and the Mann Whitney U test should be performed.

I am very happy to read this manuscript again. I think the topic is very important and should be published.

Good luck.
Sincerely
your reviewer

Author Response

Dear Reviewer

Below is our comment on your review comments.

  1. A table is suggested to characterize the sample and to understand it in a clearer way.

The content has been supplemented with a table, in which the size of the examined group was given, taking into account their affiliation to the Treatment and Control group and their gender. The table is found in verse 198.

  1. I would like to read how the questionnaire items were constructed and validated. is there a confirmatory factor analysis? The description of the instrument needs to be improved.

The study used Kendall's Concordance Coefficient W, which is a measure of concordance between several competent judges who ranked several units. It represents the ratio of the variability of the total ranks for the ranked entities to the maximum possible variability of the total ranks. The high ratio was equivalent to a consensus among the judges.

  1. It is suggested to include a specific section on statistics that justifies the choice of tests.
  2. It is recommended to perform the kolmogorov smirnov test to know if the data meet the normality assumption. If they meet the assumption of normality, it is acceptable to perform the t-test, otherwise the data should be presented as Me and IQR and the Mann Whitney U test should be performed.

We hope you find our explanation of the statistical section satisfactory. We have placed them in the manuscript at lines 266-275.

Yours faithfully,

Marcin Pasek

Reviewer 2 Report

This is an interesting study and the authors have collected unique data set. The articles is generally well written and structured. However, in my suggestion the article have some shortcoming in regards to abstract, text, method and data analysis. Below I have provided numerous remarks for the article:

  1. Need to add the issues or problem statement in the abstract.
  2. Need to clearly state the issues based on current issues regarding to the Sustainable Development Strategies.
  3. Need to clarify the Sustainable development Strategies concept.
  4. Need to add the hypotheses for the study
  5. How the reliability and validity of the data measured?
  6. The regression analysis must show in the table.
  7. The result does not show clearly the impact of environmental knowledge on implementing sustainable strategies.
  8. There no discussion on dimension of implementation sustainable strategies clearly.
  9. The literature need to cite the latest article because more the 90 percent of references are more the 3 years back.

Author Response

Dear Reviewer

Below is our comment on your review comments.

  1. Need to add the issues or problem statement in the abstract.

The problem is specified in the Abstract as a research question (lines 24-26).

  1. Need to clearly state the issues based on current issues regarding to the Sustainable Development Strategies.
  2. Need to clarify the Sustainable development Strategies concept.

These issues were introduced in the considerations that begin the Introduction. They are located in lines 38-55.

  1. Need to add the hypotheses for the study

The research hypothesis is placed in the last sentences of the Introduction (lines 193-194).

  1. How the reliability and validity of the data measured?

Relevance and reliability were tested by the authors of the tool. Due to the use of an incomplete tool in this study (the Knowledge analysis section), the indications of relevance and reliability published in the CHEAKS construction and validation article were taken as valid. It would be useful to confirm these data by examining a similar age group with the full instrument.

  1. The regression analysis must show in the table.

We put the table containing the regression analysis on line 318.

  1. The result does not show clearly the impact of environmental knowledge on implementing sustainable strategies.
  2. There no discussion on dimension of implementation sustainable strategies clearly.

We confirm that it is difficult to find direct and strong links between environmental knowledge and the implementation of sustainable strategies. We have included commentary on this issue in Discussion at lines 321-341.

  1. The literature need to cite the latest article because more the 90 percent of references are more the 3 years back.

The updated literature includes items from the last 2 years, analyzing the issue of physical activity and sustainable development policies. The 5 bibliographic items added in this connection are listed in References under numbers: 1 and 52-55.

Yours faithfully,

Marcin Pasek

Round 2

Reviewer 2 Report

Please refer to the file attached

Author Response

Dear Editor

We have carefully considered the questions addressed to us.

In addition to the brief comment below, our interpretation of the answer has been placed in the revised version of the manuscript on lines:

  1. 37-59 (Is there any problem with young person?)
  2. 213-258 (Is there any problem on outdoor physical education?, Is there any problem on environmental knowledge?, Any evidence to support the existing problem?, Any report or articles written on issue with outdoor physical education environmental knowledge and young person education system?). This section also answers the questions: “These include 46 the shortage of adequately trained teachers, unsatisfactory conditions in schools and 47 limited access to schools for children from rural areas.” Any proof for this statement? Or only argument? Or any empirical testes?
  3. This is not the hypothesis statement. Please refer to methodology style before write the hypothesis.

In lines 360-365 we initially addressed the revised hypothesis.

  1. How the authors test the validity? The author can not assume the validity. It must be tested through statistical test.

Due to the use of only a part of the tool and the size of the research sample, the authors resigned from testing the partial reliability and accuracy, because in the validation description of the Cheaks tool, the authors did not anticipate such use of the tool. If further research is conducted, an analysis of relevance, reliability, and testing of the factors included in the full scale would need to be conducted. Due to the lack of testing the psychometric value of the tool in the presented research, it should be treated as an interview questionnaire, not a test in the sense of a test with psychometric values (such as intelligence tests).

We supplemented the regression analysis with commentary (line 389-393).

  1. If, there is difficulties to find direct and strong links between environmental knowledge and the implementation of sustainable strategies, the instrument and dimension should be drop out from this study.

After reviewing the available literature, we have included a comment on this observation (line 395-408)

  1. The main references must be at least more then 50 percent of the references. It should be more.

References have been re-edited.

Kind regards,

Marcin Pasek

This manuscript is a resubmission of an earlier submission. The following is a list of the peer review reports and author responses from that submission.